psychology

climate change, communication, uncertainty, replication

**Author for correspondence:**
David Sleeth-Keppler
e-mail: david.sleeth-keppler@humboldt.edu

# Does 'When' really feel more certain than 'If'? Two failures to replicate Ballard and Lewandowsky (2015)

David Sleeth-Keppler, Stephan Lewandowsky, Timothy Ballard, Teresa A. Myers, Connie Roser-Renouf and Edward Maibach

School of Business, Humboldt State University, 1 Harpst Street, Arcata, CA 95521, USA

(iD) DS-K, 0000-0002-4195-3049

We report on two independent failures to conceptually replicate findings by Ballard & Lewandowsky (Ballard and Lewandowsky 2015 *Phil. Trans. R. Soc. A* **373**, 20140464 (doi:10.1098/rsta.2014.0464)), who showed that certainty in, and concern about, projected public health issues (e.g. impacts of climate change) depend on how uncertain information is presented. Specifically, compared to a projected range of outcomes (e.g. a global rise in temperature between 1.6°C and 2.4°C) by a certain point in time (the year 2065), Ballard & Lewandowsky (Ballard and Lewandowsky 2015 *Phil. Trans. R. Soc. A* **373**, 20140464 (doi:10.1098/rsta.2014.0464)) showed that focusing people on a certain outcome (a global rise in temperature of at least 2°C) by an uncertain time-frame (the years 2054–2083) increases certainty in the outcome, and concern about its implications. Based on two new studies that showed a null effect between the two presentation formats, however, we recommend treating the projection statements featured in these studies as equivalent, and we encourage investigators to find alternative ways to improve on existing formats to communicate uncertain information about future events.

## 1. Introduction

Scientists, policy-makers and public health organizations frequently communicate information about climate change and other public health issues in probabilistic terms. For example, to communicate the seriousness of globally rising mean temperature, communicators may use a statement that 'it is extremely likely that by 2065, average global surface temperature will rise between

**Figure 1.** Example of time-uncertain framing.

1.6°C and 2.4°C' [1]. The diffuse nature of the temperature projection, although scientifically accurate, likely interacts with the human cognitive system in undesirable ways. A general optimism bias [2,3], or motivated reasoning [4], may result in an over-focus on the lower value of the temperature range, resulting in reduced certainty the projection is real and in reduced concern. In a similar vein, Mishra *et al.* [5] demonstrated the tendency for people to use vague or ambiguous information in motivationally biased ways in the realm of consumer behaviour and performance.

To remedy the potentially negative consequences of the use of uncertain information in communicating the seriousness of climate risks, Ballard & Lewandowsky [1] devised a novel variant of the typical format that they argued to be more convincing. Instead of focusing audiences on a range of likely outcomes (e.g. 1.6–2.4°C) by a certain point in time (e.g. 2065), Ballard & Lewandowsky [1] reasoned that focusing audiences on a *certain* outcome (e.g. at least 2°C) within a range of years (e.g. 2054 and 2083) would counteract the tendency for people to engage in optimistic thinking. Ballard & Lewandowsky reasoned that communicating a certain climate outcome, albeit within an uncertain time-frame, may reduce the psychological distance of climate change in perceivers' minds [6]. Specifically, work testing construal-level theory [6] supports the notion that events that are framed as occurring in the distant future are cognitively represented in an abstract manner, relative to more proximal events. Abstract representations may reduce feelings of alarm and concern when processing information, reducing the tendency to judge risk as serious. Similarly, in addition to temporal distance, the vagueness of outcomes may undermine concern by further increasing abstraction. Thus, a focus on a certain effect expected within an uncertain time-frame, as devised by Ballard & Lewandowsky, may increase certainty and reduce the overall abstractness of a given future risk, while communicating the same underlying information as the traditional focus on a range of outcomes by a certain time (figure 1 for an example).

An experimental test of this prediction yielded positive results. Ballard & Lewandowsky found that participants who encountered a prediction of a certain outcome within an uncertain time-frame, compared to participants who encountered an uncertain outcome by a certain time, judged the climate threat as more serious, and gave stronger endorsements of the need for federal and state governments to act to mitigate climate change [1].

Given the significant policy implications of these results for communicating uncertain information about future events, we sought to submit these findings to several rigorous, conceptual replications. In study 1, we sought to conceptually replicate Ballard & Lewandowsky's [1] work by addressing methodological ambiguities in the initial study and vary the temporal focus of projections, allowing for a direct test of implications of construal-level theory [6]. Specifically, a trade-off exists between communicating a lesser outcome (for example, a relatively lower rise in global surface temperatures) at a closer point in time (for example, around 2035), and communicating a larger outcome at a more distant point in time (featured in the original study). Increasing the temporal proximity of a risk, as explained earlier, should reduce its relative abstractness, potentially increasing feelings of certainty and concern about an issue. By including this trade-off as a factor in a replication study, we tested whether relative temporal proximity to a projected outcome would increase people's certainty that the outcome is real, and concern about the issue. In study 2, we sought to extend the novel uncertainty

framing introduced by Ballard & Lewandowsy [1] to a domain of public interest unrelated to climate change, namely disease projections.

Contrary to our expectations, however, we failed to conceptually replicate Ballard & Lewandowsky's [1] initial findings, in studies conducted by two independent teams of investigators. Specifically, we found the two formats of the projection statements to yield equivalent outcomes in the climate change and disease domains. For practitioners and academics active in domains involving the public understanding (and dissemination of) scientific information, improvements in science communication methods should ideally be robust, in that they ought to reliably improve outcomes over existing methods, across domains of application and with changes made to specific instructions and informational contexts. Although the framing of an uncertain outcome as 'when, not if' has considerable intuitive appeal, we were unable to find replicable empirical support for it. We therefore encourage investigators to find alternative ways to counteract the tendency for many people to motivationally distort uncertain information about future events.

# 2. Overview of Experiment 1

The first experimental replication sought to address several methodological ambiguities in the initial Ballard & Lewandowsky [1] experiment that may have influenced the results. Specifically, Ballard & Lewandowsky dismissed respondents who failed any of four attention checks, likely resulting in a sample of respondents with an overall higher need for cognition than the average population [7]. A higher need for cognition may result in more systematic processing of information and be an unmeasured precondition for the effect. Because no further data were collected from participants who failed any of the attention checks, a test of the role of attention in obtaining the results was not possible in the original study. Secondly, the length of the text accompanying the graphs/statements communicating a certain outcome was initially somewhat shorter than the text accompanying the graphs/statements communicating a certain time projection, potentially reducing the processing burden in one condition and increasing the appeal of the novel presentation format. The length of the traditional introduction covering uncertain outcomes was around 270 words and the length of the text covering uncertain time projections around 120 words. The longer introduction covered several more examples of negative effects associated with climate change and was overall wordier without, however, substantially altering the information. The initial study also lacked a control group, reducing the ability to determine the relative magnitude of the effect.

In Experiment 1, we addressed the above concerns by including a control group, employing US units and a broader sample of respondents, including those that failed to show full attention. Additionally, we sought to increase the practical use of the novel presentation format by reducing the length of the instructions that accompanied the graphs in the initial study. Finally, we varied the temporal proximity of projections. Depending on the condition, respondents either saw a relatively closer projection in time, or a relatively distant projection, hypothesizing that proximal projections would elicit greater certainty and concern than distant projections. The distant projection statement was identical to the one featured in Ballard & Lewandowsky.

# 3. Methods (Experiment 1)

## 3.1. Participants

Five hundred and thirteen (age 18+) respondents completed the experiment using the Qualtrics$^{TM}$ platform in August of 2015. Table 1 lists relevant demographics for the sample. We recruited respondents using an online panel administered by Survey Sampling International (SSI) covering the geography of the United States. The panel provided by SSI is methodologically similar to the Qualtrics panel used in the original Ballard & Lewandowsky [1] study. Both panels employ large numbers of US adults as panel members and use propensity weighting to achieve a sample balanced to match the geography of the United States.

## 3.2. Materials

The materials were generally fashioned after the original Ballard & Lewandowsky experiment. Changes primarily addressed the methodological issues listed above, including length of instructions, measurement scales and a novel manipulation of the temporal distance of projected climate change

**Table 1.** Relevant demographics (Experiment 1) ($N = 513$).

| demographics | subgroups | statistics (%) |
|---|---|---|
| gender | male | 37.90 |
| | female | 52.40 |
| | missing | 9.50 |
| age | average | 42.75 |
| party | republican | 18.20 |
| | democrat | 38.80 |
| | other | 9.40 |
| | missing | 9.5 |
| education | less than bachelor's degree | 48.20 |
| | bachelor's degree | 27.20 |
| | higher than bachelor's degree | 15 |
| | missing | 9.70 |
| marital status | married | 46.20 |
| | single | 27.70 |
| | other | 16.70 |
| | missing | 9.50 |
| income | <$50.000 | 38.80 |
| | >=$50.000 | 51.50 |
| | missing | 9.70 |

effects. The survey was divided into three sections: training materials, manipulations and post-test measures.

Training materials included three paragraphs and questions serving as general attention checks. In essence, these questions, if answered correctly, show that participants read the passage with a high degree of attention-focus, rather than simply skimming the passages. To introduce the manipulation, the material included a graph showing the decline in the number of emperor penguins over time. The number of emperor penguins was deemed to be sufficiently dissimilar from the climate measures used for the main manipulation to avoid priming while being sufficiently similar for people to grasp the general idea of the projection formats. Depending on the condition, participants saw a graph with a certain time/uncertain decline in the population or a graph with an uncertain time/certain decline (figure 2). Following the graphs, we probed for comprehension of the graphs using two questions, requiring participants to make judgements based on information in the graphs (e.g. how likely is it that there will be *1000* breeding emperor penguin pairs in the year *2090*).

Next, participants viewed the experimental manipulations—graphs labelled 'Temperature Increases' and 'Sea-level Rise.' Each manipulation included relatively brief descriptions of the history and causes of temperature and sea-level rises. The information sections included fewer examples of causes, to reduce the burden for the respondents, but otherwise did not differ significantly from the original study. Note that Ballard & Lewandowsky's [1] original experiment included four separate projections of climate change impacts (global temperature, sea-level rise, ocean acidification and reductions in Arctic sea ice). To reduce respondent burden, we conducted our replication using only projections of global temperature and sea-level rises.

A 2 (temporal focus: distant versus close) × 2 (projection format: certainty of outcome versus certainty of time) factorial design formed the basis of the experiment. All projections were based on actual scientific estimates. Distant projections either mentioned a certain time (2065 for temperature and 2072 for sea-level rise) or a time range (2054–2083 for temperature and 2060–2093 for sea-level rise). Proximal projections included certain times of 2035 for temperature and sea-level rise, or ranges of 2027–2045. Again, depending on conditions, participants either saw uncertain effects (e.g. 'Scientists project that by 2065, global average surface temperature will rise by 2.9° to 4.3° Fahrenheit';

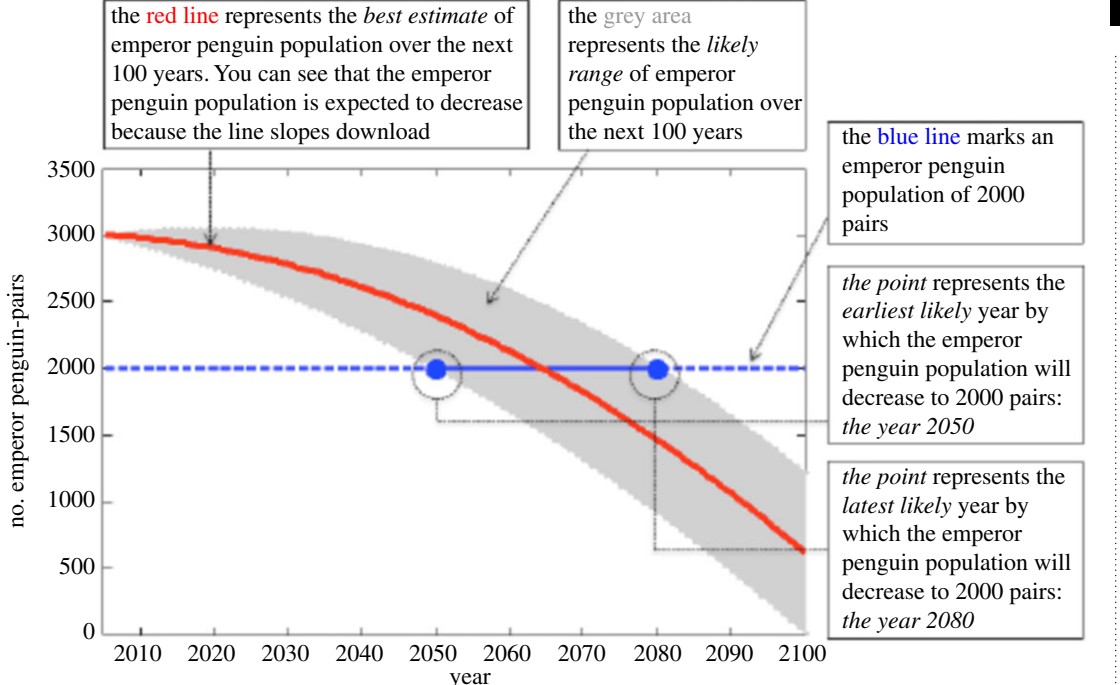

the red line represents the *best estimate* of emperor penguin population over the next 100 years. You can see that the emperor penguin population is expected to decrease because the line slopes download

the grey area represents the *likely range* of emperor penguin population over the next 100 years

the blue line marks an emperor penguin population of 2000 pairs

*the point* represents the *earliest likely* year by which the emperor penguin population will decrease to 2000 pairs: *the year 2050*

*the point* represents the *latest likely* year by which the emperor penguin population will decrease to 2000 pairs: *the year 2080*

**Figure 2.** Example of Emperor penguin training.

'Scientists project that by 2072, sea level will rise by 14 to 27 inches'), or certain effects (e.g. 'Scientists project that global average surface temperature will rise by 3.6° Fahrenheit between 2054 and 2083'; 'Scientists predict that sea levels will rise by 20 inches between 2060 and 2093'). We adjusted proximate effects to reflect reductions in magnitude (e.g. ranges of 1.25° to 2.3° Fahrenheit and 4 to 9 inches of sea-level rise) (see electronic supplementary materials). Following each manipulation, we measured respondents' perceived clarity of the figures, the effort required to comprehend them, and levels of certainty and concern about the information (each on a 9-point scale).

The post-test section featured various demographic measures, and 20 additional questions about aspects of climate change and weather events, which served as additional dependent variables. These included questions about whether climate change is happening, respondents' assessment of causes and impacts, how certain and concerned they are about climate change, and what should be done about it. An example statement measuring certainty was 'How certain are you that sea levels will rise by 8 inches between 2025 and 2045?'; an example statement measuring concern was 'How concerned are you by the above projection?'. Both measures were anchored on 9-point scales, ranging from 'not at all' to 'extremely' (see electronic supplementary materials).

## 3.3. Procedure

Respondents could complete the survey on personal computers or tablets. After indicating consent, participants read 3 passages serving as attention checks. Respondents had to check particular answer choices to show evidence of a high degree of attention. Respondents were allowed to continue, regardless of responses to these checks. Following the checks, participants were randomly assigned to the four conditions varying certainty of times and outcomes and distance of the projections. A short passage gave background on each of global temperature and sea-level rises, before the presentation of the corresponding projection statements, graphs and comprehension checks. Following the manipulation, respondents then completed post-tests and demographic measures. A fifth group, which completed only the post-test, served as an experimental control group.

# 4. Results (Experiment 1)

35.8% of respondents passed all three of the attention checks included at the beginning of the study (50.2% passed check 1, 42.8% passed check 2 and 66.7% passed check 3), and 36.3% passed both comprehension checks included in the manipulation, regarding the penguin population. In the

**Table 2.** Results from multiple regression analyses (Experiment 1). Note: values are standardized coefficients.

| | certainty | concern |
|---|---|---|
| gender | −0.03 | −0.043 |
| age | −0.133* | −0.092 |
| race | −0.062 | −0.073 |
| education | 0.067 | 0.021 |
| income | −0.001 | 0.03 |
| ideology | −0.077 | −0.241*** |
| party ID | −0.261*** | −0.216*** |
| adj. $R^2$ | 9.10% | 16.30% |
| attn checks | −0.139* | 0.006 |
| total time | 0.014 | 0.081 |
| penguin Qs | −0.093 | −0.117 |
| effort | 0.005 | −0.077 |
| clarity | 0.365*** | 0.246*** |
| adj. $R^2$ | 22.70% | 23.70% |
| focus | 0.012 | −0.052 |
| timing | 0.078 | −0.071 |
| interaction | −0.18 | 0.016 |
| adj. $R^2$ | 23.60% | 24.20% |

*$p < 0.05$; **$p < 0.01$; ***$p < 0.001$.

original Ballard & Lewandowsky [1] experiment, 45 of 234 (19.2%) qualified participants failed any of four separate attention checks, suggesting our participants in the present study showed relatively less attention. We inquired with SSI, our sample provider for Experiment 1, about the quality of the data. SSI's internal research shows that some or all of our three initial attention checks are easily missed by respondents, who nonetheless subsequently show an adequate level of attention for most survey purposes [8]. Supporting this notion is our supposition that the attention filters employed by Ballard & Lewandowsky in their original experiment required relatively less involvement and focus. Compared to the original Ballard & Lewandowsky experiment, who used relatively simple multiple-choice questions without introductory text (e.g. What is $2 + 3$?), our attention checks required active reading of instructions. Relatedly, participants in online studies tend to read in an 'F' pattern, rather than reading every sentence, suggesting a lighter attention-focus. Thus, it appears that the nature of the attention filters used in Experiment 1, compared to the original experiment, contribute to the high attention-focus failure-rate of Experiment 1. Crucially for the present purpose, two multiple regression analyses, regressing the number of correct responses to the attention checks and penguin comprehension questions as separate indicators on the two response variables, (a) certainty that the climate projections will occur and, (b) concern about the projections, revealed nonsignificant or opposite patterns (table 2). Attention checks showed a negative coefficient beta for certainty ($b = -0.139$, $p < 0.05$) and a null effect for concern ($b = 0.006$); the penguin measure a null effect for both (b_concern $= -0.093$; b_worry $= -0.117$, both n.s.).

As the multiple regression results in table 2 reveal, neither the independent effect of temporal distance (focus), nor the effect of projection format (varying certainty of outcomes or times), nor their interaction reached statistical significance.

To replicate more directly the original conditions of Ballard & Lewandowsky [1], we repeated the analysis with only those respondents included who passed all three initial attention checks. Table 3 shows means and ANOVA results for this restricted subset of participants, again revealing null results.[1]

---

[1]To further assess the extent to which attention influenced the effects, we tested whether the amount of attention paid moderated the effect of focus, timing, or the focus by timing interaction. In no case did attention significant interact with any of these treatment effects.

**Table 3.** Effects of focus and projection format on certainty and concern about climate projections on a base of respondents who passed all 3 attention checks (Experiment 1). Note: values are means on a 9-point scale.

| focus: | distant projection | | proximal projection | | F | | | |
|---|---|---|---|---|---|---|---|---|
| timing: | certain time | certain outcome | certain time | certain outcome | focus | timing | focus × timing | N |
| certainty that projections will occur | 6.02 | 5.64 | 5.74 | 4.94 | 1.03 | 1.5 | 0.19 | 111 |
| concern about the projections | 5.86 | 5.86 | 6.33 | 5.88 | 0.27 | 0.21 | 0.22 | 111 |

**Table 4.** Bayes Factors (and their reciprocals) for a Bayesian regression analysis involving the same predictors and dependent variables as the frequentist analysis reported in table 2. Note: Bayes Factors (BF) greater than 1 provide evidence for an effect. Strength of evidence is considered merely anecdotal (1–3); moderate (3–10); strong (10–30); very strong (30–100); or decisive (>100).

| | certainty | concern |
|---|---|---|
| gender | 0.17 (6.04) | 0.13 (7.50) |
| age | 55.85 | 8.70 |
| race | 0.28 (3.54) | 0.26 (3.83) |
| education | 0.12 (7.97) | 0.13 (7.99) |
| income | 0.13 (7.97) | 0.18 (5.43) |
| ideology | 240.89 | 1 039 175 052 |
| party ID | 362 068 | 49 182 081 |
| attn checks | 0.49 (2.04) | 0.17 (5.90) |
| total time | 0.67 (1.48) | 0.13 (7.86) |
| penguin Qs | 0.25 (4.06) | 0.20 (5.11) |
| effort | 0.14 (7.25) | 0.16 (6.25) |
| clarity | 684 439 | 31.99 |
| focus | 0.42 (2.40) | 0.13 (7.88) |
| timing | 3.07 | 1.53 |
| interaction | 0.13 (7.80) | 0.12 (8.09) |

Because frequentist statistics are notorious for their inability to provide support for the *absence* of an effect, we also conducted a Bayesian analysis using the linear-models functions (generalTestBF and lmBF) in the BayesFactor package [9] in R. Unlike frequentist techniques, Bayesian statistics permit comparisons between *any* pair of statistical models, including 'null' models for the absence of effects. Table 4 shows the results of a Bayesian regression analysis that parallels the frequentist model in table 2 (with records containing missing values deleted; $N = 319$). The table entries are Bayes Factors associated with each predictor tested against the null model (intercept only).

Any Bayes factor (BF) greater than 1 provides evidence in favour of an effect, with values in the range 1–3 considered merely anecdotal, 3–10 considered moderate evidence and anything above 10 considered strong (10–30), very strong (30–100) or decisive (greater than 100) [10]. Table 3 shows that there is at least strong evidence for an effect of Ideology, Party ID and Clarity on both outcome variables. For Certainty, there is an additional strong effect of Age.

Bayes Factors less than one are considered to provide evidence for the *absence* of an effect, and their reciprocal (1/BF) is interpretable along the same scale as any BF greater than 1. To illustrate, a BF= 0.1 provides strong evidence (1/0.1 = 10) for the absence of an effect, and a BF = 0.01 provides decisive

(1/0.01 = 100) evidence for the absence of an effect. Applying this interpretation to table 3 implies that there was at least anecdotal evidence against the effect of the main experimental variable of interest, Focus.

To further establish whether any of the experimental variables or their interaction had an effect, we compared the full model for each dependent variable (i.e. including all predictors in table 3) against a reduced model that omitted only the experimental variables (Focus and Timing) and their interaction. The model comparison revealed anecdotal evidence for the absence of any experimental effects for Certainty (reciprocal BF = 2.70) and very strong evidence of the absence for Concern (42.89).

## 5. Discussion (Experiment 1)

The present study addressed an important ambiguity in Ballard & Lewandowsky's [1] initial experiment about the role of attention-focus (or generally, the need for cognition) to obtain their results. The inclusion of an attention-check variable in our conceptual replication allowed for a direct test of the hypothesis that an increased concern about climate-related negative outcomes depends on sufficient attention-focus to process the materials. The regression analyses shown in table 2 do not support this notion, nor the initial finding that uncertainty frames involving a certain outcome by an uncertain time yield more concern about climate change than do uncertainty frames involving uncertain outcomes, yet certain times. The analysis, on a base of the most attentive participants (those who passed all three attention checks), did not yield the effect, even though participants should have been highly motivated to process the graphs and accompanying information. Future research could examine whether the need for cognition or other motivational variables may affect the processing of the uncertainty information and the framing of the future projections to increase/decrease concern about public interest issues.

The failure to replicate Ballard & Lewandowsky's [1] original findings raises important questions about the viability of the outcome-certain/time-uncertain future projection format as a communication tool. Even though failing to replicate a result, especially when conducting a conceptual replication, does not necessarily mean the underlying effect is not real, a close replication failure does suggest the effect may be fragile, and crucially, may not function better as a practical device in public interest communications, compared to traditional projection formats.

The present replication also included several simplifications of the instructions, which should have facilitated a replication, and a theoretical extension of the effect to a more proximal time-frame, directly testing temporal construal theory. Proximity, in particular, should have facilitated a replication of the original results, because more proximate threats are more concrete, and therefore, more difficult to ignore. Our results do not support this notion, perhaps because the proximal projections still did not appear threatening.

Unlike in the original study, our current replication included only two climate-related impacts (temperature and sea-level rises), whereas the original experiment included four. It is possible that participants matured over the course of the original study, and that exposure to relatively more projection statements facilitates the processing of the information, compared to exposure to fewer statements. Additional analysis of the present replication results suggests that this may be the case. The perceived clarity of the figures increased significantly with the second impact, and effort needed decreased significantly, suggesting that respondents improved their ability to interpret the statements and figures. Given the possibility of maturation, and to submit the original Ballard & Lewandowsky [1] results to yet another conceptual replication, we conducted a second experiment. Table 5 summarizes various key differences between the present Experiment 1 and the Ballard & Lewandowsky [1] experiment.

The second experiment examined the variation of outcome (un)certainty versus time (un)certainty in the context of fictitious diseases, extending the effect to another domain of public health. This experiment had been in the planning stages and pretested and even though the findings in Experiment 1 were negative, we decided to move forward with a domain change. In principle, communication about uncertainty using the 'when, not if' method should increase concern about an issue independently of context, similarly to how various graphing tools are context-free. To test whether the nature of the uncertainty statements can be obtained without graphical information, which may not always be feasible to use for communications (e.g. radio, conversations, podcasts), we decided to omit the graphs from the manipulation in Experiment 2 and replaced them with verbal instructions about the projection format. Finally, the second experiment did not feature a training task, because no unfamiliar graphical information was presented to participants, compared to Experiment 1.

**Table 5.** Major differences between Experiment 1 and Ballard & Lewandowsky [1].

| | Experiment 1 | Ballard & Lewandowsky [1] |
|---|---|---|
| sample provider | SSI | Qualtrics |
| climate indicators used | temperature; sea-level rise; ocean acidification; reductions in Arctic sea ice | temperature; sea-level rise |
| no. of attention checks | 3 | 4 |
| total recruited N | 513 | 324 |
| N passing all attention checks | 184 | 189 |
| likely involvement required to pass attention checks | relatively high (focus on reading instructions) | relatively low (easy answers, low effort) |
| missing values | treated with list-wise deletion | treated with 'hot-deck' imputation in the R package |

# 6. Methods (Experiment 2)

The method, ethics approval from the University of Bristol, and analysis plan were preregistered and are available at https://osf.io/pcebv/. The preregistration contains an exact copy of all conditions of the online survey. Any deviations from the preregistered method and analysis plan are explicitly noted below.

## 6.1. Participants

We contracted Qualtrics.com during May 2016 to collect 100 complete respondents for the online experiment. Qualtrics(TM) administers Internet surveys to representative samples, and the current pool of respondents was drawn from the US geography using propensity sampling from a large panel of residents. Owing to a rapid influx of respondents, Qualtrics returned a larger-than-contracted sample of 223 completed responses which we used for the analysis. Each respondent had passed two attention filter questions. Three participants who did not provide final consent at the end of the survey were removed, yielding a final sample of 220 observations.

## 6.2. Materials

We divided the survey into five sections that each addressed one of the following fictitious diseases: Laerosis, Ralinosis, Hilenfia syndrome, Gearn's disease and Cerioa. Similar to Experiment 1, each section began with a paragraph providing fictitious information about the relevant disease. At the end of each paragraph, a statement was presented that described the expected future trends in the disease's prevalence. For each disease, one of two statements was presented, chosen at random for each respondent. One statement expressed the range in possible outcomes that would occur by a given future time point.

For example:

3 out of every 100 adults in Africa will be infected with Laerosis this year. We know with high confidence that the number of Laerosis cases will increase by 2027, the only question is by how much. Current projections are that by that time between 9 and 14 out of every 100 adults in Africa will be infected with Laerosis.

The other statement expressed the range in possible timeframes within which a given outcome would occur. For example:

3 out of every 100 adults in Africa will be infected with Laerosis this year. We know with high confidence that the number of Laerosis cases will increase to 12 out of every 100 adults in Africa, the only question is when. Current projections are for this figure to be reached between 2025 and 2030.

As in Experiment 1, the two possible statements always reflected the same fictitious trend, so they were equivalent with respect to the severity of the underlying projection.

## 6.3. Procedure

After reading an information screen and providing initial consent, participants completed the five sections in a random order. In each section, the statement that was presented (i.e. outcome uncertainty versus time

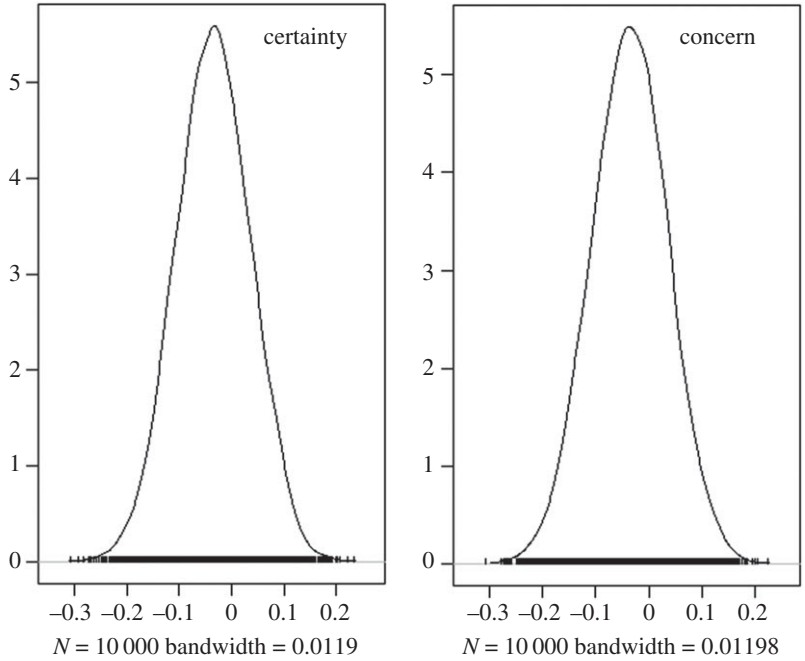

**Figure 3.** Posterior densities for certainty and concern (Experiment 2).

uncertainty) was randomly determined, reflecting a within-subjects design. After reading the relevant paragraph and statement, we asked respondents how certain they were that the disease prevalence will change as projected, and how concerned they were by the projection using a 9-point scale from $1 =$ Not at all certain/concerned to $9 =$ Extremely certain/concerned. Several choice items followed that queried whether the projected change will be good or bad for the population. After completion of the survey, participants indicated their age and gender before providing final consent.

# 7. Results (Experiment 2)

## 7.1. Data pre-processing

One respondent inexplicably responded to *both* uncertainty formats for the same disease, suggesting a failure of randomization, and was therefore excluded from the analysis. The remaining 219 responses included one participant whose reported age was 100 (the maximum value of the scale used for a slider to report age) and a further four participants whose completion times fell more than 3 s.d. above the mean of 711.7 s. Those outlying observations were also removed, yielding a final set of 214 participants for analysis. None of the principal results are materially affected if those observations are retained.

## 7.2. Preregistered analyses

In line with the preregistered analysis plan, we first examined how the two projection formats affected the certainty and concern items in a within-subjects analysis, averaging across the diseases within each projection format. For the certainty item, mean ratings (on a scale from 1 to 9) were 5.81 and 5.76 for the time-uncertain and outcome-uncertain formats, respectively. For the concern item, the mean ratings were 5.78 and 5.73 for the time-uncertain and outcome-uncertain formats, respectively. We used Bayesian paired-sample (within-subject) *t*-tests for this comparison, using the Bayes Factor package [9] in R. The Bayes factors *in favour of the null hypothesis* were 11.11 and 10.97, respectively, for the certainty and concern items. The experiment therefore returned strong evidence for the null hypothesis of no effect of the projection format. Figure 3 illustrates the results by showing the posterior densities (based on 10 000 MCMC samples) of the difference between projection formats.

To comply with the preregistered analysis plan, we next conducted five independent between-subject Bayesian *t*-tests for the five diseases separately. The tests provided at least moderate evidence for the null hypothesis in each case, with the smallest BF = 4.24 for Laerosis for the certainty item, and the smallest BF = 3.33 for concern for Hilenfia syndrome.

We cross-tabulated the number of responses for the categorical choice item ('Do you think the rise in the number of people with [disease X...] will be good, bad, or neither good nor bad'?) and found no differences between the projection formats that would have warranted further exploration. For the time-uncertain format, 49 responses were 'good', and 384 'bad', with a further 93 undecided. For the outcome-uncertain format, the distribution of responses was 52, 393, and 99 for 'good', 'bad' and undecided, respectively ($\chi^2(2) = 0.08$, $p > 0.10$).

## 7.3. Additional analyses

The randomization of projection formats implied that some participants ($N = 17$) received all diseases in the same projection format. Those participants necessarily contributed missing observations to the preregistered within-subject comparisons. We therefore additionally explored a mixed-effect model using the *lmer* function in R. This model considered all observations and included fixed effects of projection format and, fully crossed with that design factor, three further measures of the statistics presented for each disease; namely, the time range in the time-uncertain condition (e.g. 6 years for Laerosis; see method section), the time until the projection in the outcome-uncertain condition (e.g. 12 years for Laerosis, using 2015 as the present), the current value of the variable (e.g. 3 for Laerosis), and the increment from that current value to the projected value in the time-uncertain format (e.g. 9 for Laerosis). The model additionally included a random intercept and a random effect of the projection format for each participant.

None of the coefficients were found to be significant for the certainty item (largest absolute $t = -1.46$ for time until projection) or the concern item (largest absolute $t = 1.20$ for the increment of disease incidence). We additionally ran the same model on the serious item (How serious of a problem do you think the increase in the number of people with [disease X] will be...), and found that for this model the effect of increment was significant, $t = 2.30$, $p < 0.05$, suggesting that people's perceptions of the seriousness of a disease were calibrated to the numerical magnitude of the future increment in the number of projected patients.

# 8. Discussion (Experiment 2)

Experiment 2, a conceptual replication of Experiment 1 using a series of fictitious diseases, again failed to show evidence that a framing of future outcomes as time-uncertain, rather than outcome-uncertain, provides an advantage over the opposite framing method. Similar to the result in Experiment 1, participants who encountered a time-uncertain, outcome-certain framing about the development of various diseases showed no difference in the certainty that the information is real and concern for the diseases, compared to participants who encountered a time-certain, outcome-uncertain framing about the same diseases.

# 9. General discussion

Ballard & Lewandowsky's [1] 'when, not if' framing of projections represents an intuitively powerful way to present data to increase feelings of certainty and concern about a wide range of public interest issues, without changing the underlying data. The initial support for the effectiveness of 'when, not if', compared to the traditional 'if, not when', was therefore, unsurprising.

We embarked on the replication studies reported in this article with much confidence that we would replicate the initial results while improving minor methodological concerns and extending the generality of the initial finding. The inability to replicate the general effect in Experiment 1 led us to refine the procedure and switch to a verbal presentation format that we expected to accentuate the difference between the 'when, not if' and 'if, not when' framings. Those efforts were supported by several pilot studies using small sample sizes and conducted in the laboratory, that in the lead-up to Experiment 2 again showed results favouring the 'when, not if' framing. We conducted the pretests largely to examine the clarity of the instructions and materials and conclude that the statistical results of the pilot studies are random noise. The fact the effect ultimately did not replicate in two studies with large numbers of respondents in a generally representative survey, including an experiment employing a preregistered protocol, significantly reduces our confidence in the existence of the effect.

We suggest the following implications for public communicators. Even though we cannot empirically support the notion that the 'when, not if' framing represents an improvement over traditional projection

formats, we can state confidently that the reframing *does not harm* public communication about climate, health, and potentially other issues in the public domain. Practitioners may therefore replace traditional methods of projecting outcomes with the 'when, not if' framing whenever they feel the novel presentation format may be appropriate.

## 10. Conclusion

Public support for issues in the public interest may depend on how uncertain information is presented. We describe studies failing to show improvement of a novel presentation method, but encourage practitioners to exercise their best judgement in which format to use in their communication attempts.

Research ethics. To carry out the research protocol for Experiment 1, we obtained approval from the George Mason University (Virginia, USA) Research Development, Integrity and Assurance Office. To carry out the research protocol for Experiment 2, we received a favourable ethical opinion from the Faculty of Science Human Research Ethics committee at the University of Bristol (United Kingdom).

Animal ethics. We were not required to complete an ethical assessment prior to conducting your research.

Permission to carry out fieldwork. No permissions were required prior to conducting this research.

Data accessibility. The data for both experiments are available in the Dryad repository: https://doi.org/10.5061/dryad.74bf15t [11].

Authors' contributions. D.S.-K. made a substantial contributions to the conception and design of Experiment 1 and analysed and interpreted the data. He also wrote the first draft of the manuscript. S.L. and T.B. conceived and designed Experiment 2 and analysed and interpreted the data. S.L. also made substantial revisions of the first manuscript. C.R.-R. conceived and designed Experiment 1, acquired the data and analysed and interpreted the results and made intellectual contributions to the revisions of the manuscript. T.M. made substantial contribution to the conception and design of Experiment 1, interpreted the data and made revisions to the manuscript. E.M. substantially contributed to the conception and design of Experiment 1, interpreted the data and made revisions to the manuscript. All authors read and approved the manuscript before submission.

Competing interests. We have no competing interests.

Funding. The research was funded by a grant from the National Aeronautics and Space Administration, and by a Wolfson Fellowship from the Royal Society awarded to S.L.

Acknowledgements. We thank Louise Kenney for her support in the design and implementation of Experiment 2.

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
