## [Reviewer comments · Royal Society Open Science]

Review History

RSOS-180475.R0 (Original submission)

Review form: Reviewer 1

Is the manuscript scientifically sound in its present form?

Yes

Are the interpretations and conclusions justified by the results?

Yes

Is the language acceptable?

Yes

Is it clear how to access all supporting data?

Yes

Do you have any ethical concerns with this paper?

No

Have you any concerns about statistical analyses in this paper?

No

Recommendation?

Major revision is needed (please make suggestions in comments)

Comments to the Author(s)

I read this manuscript with great interest. It attempted to replicate the findings of a previous paper (Ballard & Lewandowsky) that audiences are more responsive to messages that express uncertainty in predictions of the future through a range of time (and a certain outcome) rather than a range of outcomes (and a certain time frame). I reviewed this manuscript as well as the original paper (Ballard & Lewandowsky). I applaud the authors for carefully writing up the findings of their studies though they found null results; I think this adds to a very beneficial trend in the field. In addition, the authors took care to incorporate a variety of analytic approaches in their manuscript (e.g., Bayes Factors, mixed models), which adds to the strength of the manuscript.

However, I felt that the authors could strengthen the paper by addressing and/or clarifying several points. The studies that the authors conducted in this manuscript differed from the original Ballard & Lewandowsky study in several ways that might have affected the results, and I thought that each of these possibilities might be discussed and/or investigated more thoroughly. Generally, I thought a bit more time could have been spent digging into the differences between these studies and Ballard & Lewandowsky, and that doing so could help to make the implications of this research clearer. Below, please find my comments to the authors. I hope that they find them helpful and I hope to see this manuscript in print in the future.

First, I wondered whether the authors could in fact test some of their underlying assumptions for the differences between these past studies by re-analyzing existing data and/or conducting new studies. For example, the authors mention on pg. 3 that the Ballard & Lewandowsky study had omitted participants who failed one of 5 attention checks, which could have behind a group of participants with greater need for cognition and who would thus be more likely to exhibit effects. The authors could perhaps go back to the original data to see if there are any effects of the uncertainty framing when these participants who failed the attention check are included, that is if any further data were collected from these participants (the phrase “dismissed” on pg. 3 seems to imply that no data were collected from these Qualtrics panel participants, but the authors might clarify this point). Or, the authors could use whether participants failed the attention checks or not (e.g., 1=passed, 0=did not pass) as a moderator of the effects (in either the replication studies or the original study) in order to test whether attention checks serve as a proxy for some necessary precondition for the beneficial effects of the uncertainty framing. Or, in a replication study, the authors could directly include a measure of need for cognition as a possible moderator of effects. I think addressing this point about the attention checks is especially important because the authors discuss the aspect of highly motivated audiences in their conclusions (see pg. 9, “These situations may include communication to highly motivated audiences, with a high degree of investment in the issue, who may feel the vagueness of traditional outcome statements undermines support for the issue at hand”), or else the authors might want to revise this conclusion since there does not appear to be any evidence for it. On a related note, I wondered if whether participants successfully passed the training items might interact with the uncertainty treatments – perhaps successful training/comprehension is a necessary precondition for effects?

Second, the authors mentioned that they removed part of the instructions to keep the word count consistent between conditions. I wondered what exactly the authors omitted from the instructions (e.g., the average projection estimate that was included in Ballard & Lewandowsky)?

I appreciate the drive to keep the word count consistent between conditions, but I also wondered whether there was anything critical in these instructions that might have prompted the differences between studies. For example, could it be that including the average projection in the original studies had some kind of deleterious effect (e.g., including the average projection introduced even more uncertainty and undermined the outcome-uncertain condition compared to the time-uncertain condition in Ballard & Lewandowsky), and once removed there are no differences between conditions?

Third, I wondered about the attention checks included in the current manuscript. Did they differ from the attention checks in the original Ballard & Lewandowsky study? From my reading, it appears a lot more participants (64.2%) failed at least one of the attention checks used in the current manuscript than those used in the original Ballard & Lewandowsky study (45 of 241 participants, 18.7%). Were these more stringent attention checks in some way, and how should readers understand these differences and how they might affect the results?

Fourth, in the original study in Ballard & Lewandowsky, the authors mention additional methodological steps that were undertaken (e.g., was multiple imputation, propensity weighting invitations). Were these procedures used in the same way in the current study, and why or why not? How do these factors affect the original results (e.g., when multiple imputation is not used, controlling for demographics vs. not)? This kind of line of inquiry (along with re-analyzing the original data from Ballard & Lewandowsky including participants who failed attention checks, if possible) is one way that the authors could help to indicate whether the original results in Ballard & Lewandowsky seem to be fragile or robust, so that audiences can weigh the strength of the original evidence alongside the replications.

Fifth, I wondered whether the within-subjects design of Experiment 2 could have undermined the results, given that participants were seeing statements about diseases one right after the other with different kinds of uncertainty. Could this transition between different kinds of uncertainty have been confusing for participants, especially since they were not provided any kind of training in this study (as opposed to Ballard & Lewandowsky and Experiment 1)?

Sixth, I was also curious why the choice was made to move from sea level rise to public health when the original effect on sea level rise had not yet been replicated. The authors might discuss this choice, and the differences between these contexts, more in detail.

Finally, the authors also mentioned that they did find further evidence for the original Ballard & Lewandowsky effects in pilot testing (pg. 9). What was different about the lab studies and pilot testing? Are there any insights here, or does it just appear to be random noise?

Review form: Reviewer 2

Is the manuscript scientifically sound in its present form?

No

Are the interpretations and conclusions justified by the results?

No

Is the language acceptable?

Yes

Is it clear how to access all supporting data?

Yes

Do you have any ethical concerns with this paper?

No

Have you any concerns about statistical analyses in this paper?

Yes

Recommendation?

Major revision is needed (please make suggestions in comments)

Comments to the Author(s)

Two failures to replicate Ballard and Lewandowsky

The manuscript reports two studies that include provide a conceptual replication of an article about presentation of uncertain information and consequences for decision making. My concerns and suggestions to address them are listed below.

1. Replications are clearly an important part of modern scientific approaches, and the results of those replication attempts deserve publication in some format. I had several issues with the present report, but some of these concerns could be ameliorated if this type of publication is consistent with the mission of the journal.
2. It is not clear that the original results have had an impact that would warrant publication of a replication. The original has been cited ten times, and the authors on the replication appear to also be the authors on the original paper.
3. The replication is not a direct replication of the original studies – multiple factors are changed, including the design and the analytic approach. This makes it very difficult to compare across manuscripts, or to determine why the replication might have failed. Was the first report a collection of type I errors, or were there other important methodological or analytic differences that resulted in a failure. Further investigating the reason for failure could increase the contribution of the manuscript. At the very least, including comparable information in the replication would permit readers to compare across studies.

Decision letter (RSOS-180475.R0)

12-Oct-2018

Dear Dr Sleeth-Keppler,

The editors assigned to your paper ("Does "When" Really Feel More Certain than "If"?: Two failures to replicate Ballard and Lewandowsky (2015)") have now received comments from reviewers. We would like you to revise your paper in accordance with the referee and Associate Editor suggestions which can be found below (not including confidential reports to the Editor). Please note this decision does not guarantee eventual acceptance.

In particular, please address the comments of reviewer 1 in detail. This should also cover the more general points from reviewer 2.

Please submit a copy of your revised paper before 04-Nov-2018. Please note that the revision

deadline will expire at 00.00am on this date. If we do not hear from you within this time then it will be assumed that the paper has been withdrawn. In exceptional circumstances, extensions may be possible if agreed with the Editorial Office in advance. We do not allow multiple rounds of revision so we urge you to make every effort to fully address all of the comments at this stage. If deemed necessary by the Editors, your manuscript will be sent back to one or more of the original reviewers for assessment. If the original reviewers are not available, we may invite new reviewers.

- Data accessibility

<http://datadryad.org/submit?journalID=RSOS&manu=RSOS-180475>

- Competing interests

- Authors' contributions

All submissions, other than those with a single author, must include an Authors' Contributions section which individually lists the specific contribution of each author. The list of Authors should meet all of the following criteria; 1) substantial contributions to conception and design, or

acquisition of data, or analysis and interpretation of data; 2) drafting the article or revising it critically for important intellectual content; and 3) final approval of the version to be published.

- Acknowledgements

- Funding statement

Please note that Royal Society Open Science charge article processing charges for all new submissions that are accepted for publication. Charges will also apply to papers transferred to Royal Society Open Science from other Royal Society Publishing journals, as well as papers submitted as part of our collaboration with the Royal Society of Chemistry (<http://rsos.royalsocietypublishing.org/chemistry>). If your manuscript is newly submitted and subsequently accepted for publication, you will be asked to pay the article processing charge, unless you request a waiver and this is approved by Royal Society Publishing. You can find out more about the charges at <http://rsos.royalsocietypublishing.org/page/charges>. Should you have any queries, please contact openscience@royalsociety.org.

on behalf of Dr Anastasia Christakou (Associate Editor) and Prof. Antonia Hamilton (Subject Editor)
openscience@royalsociety.org

Comments to Author:

Reviewers' Comments to Author:

Reviewer: 1

Comments to the Author(s)

I read this manuscript with great interest. It attempted to replicate the findings of a previous paper (Ballard & Lewandowsky) that audiences are more responsive to messages that express uncertainty in predictions of the future through a range of time (and a certain outcome) rather

than a range of outcomes (and a certain time frame). I reviewed this manuscript as well as the original paper (Ballard & Lewandowsky). I applaud the authors for carefully writing up the findings of their studies though they found null results; I think this adds to a very beneficial trend in the field. In addition, the authors took care to incorporate a variety of analytic approaches in their manuscript (e.g., Bayes Factors, mixed models), which adds to the strength of the manuscript.

However, I felt that the authors could strengthen the paper by addressing and/or clarifying several points. The studies that the authors conducted in this manuscript differed from the original Ballard & Lewandowsky study in several ways that might have affected the results, and I thought that each of these possibilities might be discussed and/or investigated more thoroughly. Generally, I thought a bit more time could have been spent digging into the differences between these studies and Ballard & Lewandowsky, and that doing so could help to make the implications of this research clearer. Below, please find my comments to the authors. I hope that they find them helpful and I hope to see this manuscript in print in the future.

First, I wondered whether the authors could in fact test some of their underlying assumptions for the differences between these past studies by re-analyzing existing data and/or conducting new studies. For example, the authors mention on pg. 3 that the Ballard & Lewandowsky study had omitted participants who failed one of 5 attention checks, which could have behind a group of participants with greater need for cognition and who would thus be more likely to exhibit effects. The authors could perhaps go back to the original data to see if there are any effects of the uncertainty framing when these participants who failed the attention check are included, that is if any further data were collected from these participants (the phrase “dismissed” on pg. 3 seems to imply that no data were collected from these Qualtrics panel participants, but the authors might clarify this point). Or, the authors could use whether participants failed the attention checks or not (e.g., 1=passed, 0=did not pass) as a moderator of the effects (in either the replication studies or the original study) in order to test whether attention checks serve as a proxy for some necessary precondition for the beneficial effects of the uncertainty framing. Or, in a replication study, the authors could directly include a measure of need for cognition as a possible moderator of effects. I think addressing this point about the attention checks is especially important because the authors discuss the aspect of highly motivated audiences in their conclusions (see pg. 9, “These situations may include communication to highly motivated audiences, with a high degree of investment in the issue, who may feel the vagueness of traditional outcome statements undermines support for the issue at hand”), or else the authors might want to revise this conclusion since there does not appear to be any evidence for it. On a related note, I wondered if whether participants successfully passed the training items might interact with the uncertainty treatments - perhaps successful training/comprehension is a necessary precondition for effects?

Second, the authors mentioned that they removed part of the instructions to keep the word count consistent between conditions. I wondered what exactly the authors omitted from the instructions (e.g., the average projection estimate that was included in Ballard & Lewandowsky)? I appreciate the drive to keep the word count consistent between conditions, but I also wondered whether there was anything critical in these instructions that might have prompted the differences between studies. For example, could it be that including the average projection in the original studies had some kind of deleterious effect (e.g., including the average projection introduced even more uncertainty and undermined the outcome-uncertain condition compared to the time-uncertain condition in Ballard & Lewandowsky), and once removed there are no differences between conditions?

Third, I wondered about the attention checks included in the current manuscript. Did they differ from the attention checks in the original Ballard & Lewandowsky study? From my reading, it appears a lot more participants (64.2%) failed at least one of the attention checks used in the

current manuscript than those used in the original Ballard & Lewandowsky study (45 of 241 participants, 18.7%). Were these more stringent attention checks in some way, and how should readers understand these differences and how they might affect the results?

Fourth, in the original study in Ballard & Lewandowsky, the authors mention additional methodological steps that were undertaken (e.g., was multiple imputation, propensity weighting invitations). Were these procedures used in the same way in the current study, and why or why not? How do these factors affect the original results (e.g., when multiple imputation is not used, controlling for demographics vs. not)? This kind of line of inquiry (along with re-analyzing the original data from Ballard & Lewandowsky including participants who failed attention checks, if possible) is one way that the authors could help to indicate whether the original results in Ballard & Lewandowsky seem to be fragile or robust, so that audiences can weigh the strength of the original evidence alongside the replications.

Fifth, I wondered whether the within-subjects design of Experiment 2 could have undermined the results, given that participants were seeing statements about diseases one right after the other with different kinds of uncertainty. Could this transition between different kinds of uncertainty have been confusing for participants, especially since they were not provided any kind of training in this study (as opposed to Ballard & Lewandowsky and Experiment 1)?

Sixth, I was also curious why the choice was made to move from sea level rise to public health when the original effect on sea level rise had not yet been replicated. The authors might discuss this choice, and the differences between these contexts, more in detail.

Finally, the authors also mentioned that they did find further evidence for the original Ballard & Lewandowsky effects in pilot testing (pg. 9). What was different about the lab studies and pilot testing? Are there any insights here, or does it just appear to be random noise?

Reviewer: 2

Comments to the Author(s)

Two failures to replicate Ballard and Lewandowsky

The manuscript reports two studies that include provide a conceptual replication of an article about presentation of uncertain information and consequences for decision making. My concerns and suggestions to address them are listed below.

1. Replications are clearly an important part of modern scientific approaches, and the results of those replication attempts deserve publication in some format. I had several issues with the present report, but some of these concerns could be ameliorated if this type of publication is consistent with the mission of the journal.
2. It is not clear that the original results have had an impact that would warrant publication of a replication. The original has been cited ten times, and the authors on the replication appear to also be the authors on the original paper.
3. The replication is not a direct replication of the original studies – multiple factors are changed, including the design and the analytic approach. This makes it very difficult to compare across manuscripts, or to determine why the replication might have failed. Was the first report a collection of type I errors, or were there other important methodological or analytic differences that resulted in a failure. Further investigating the reason for failure could increase the contribution of the manuscript. At the very least, including comparable information in the replication would permit readers to compare across studies.

Author's Response to Decision Letter for (RSOS-180475.R0)

See Appendix A.

RSOS-180475.R1 (Revision)

Review form: Reviewer 1

Is the manuscript scientifically sound in its present form?

Yes

Are the interpretations and conclusions justified by the results?

Yes

Is the language acceptable?

Yes

Is it clear how to access all supporting data?

No

Do you have any ethical concerns with this paper?

No

Have you any concerns about statistical analyses in this paper?

No

Recommendation?

Major revision is needed (please make suggestions in comments)

Comments to the Author(s)

I have read the revised manuscript as well as the authors' responses and would like to take the opportunity to clarify a few of my remarks to the authors and to ask some outstanding questions. I include the authors' comments and my responses to their comments below.

Comment #1: Regarding Reviewer 1's first set of comments, we clarified in the Overview to Exp. 1 that no further data was collected from Ballard and Lewandowsky's original dismissed participants, making any further tests of attention as a moderator of the effects impossible. This was also a big motivation for us to conduct the present studies. Collecting more data, as Reviewer 1 suggests, would defeat the purpose of diagnosing the role of attention further, because Experiment 1 in the present manuscript was already designed to address this issue. Our original manuscript included analyses of the attention checks as variables in a multiple regression analysis (finding null results), and we also replicated the conditions of the original Ballard and Lewandowsky (2015) study by analyzing our present Experiment 1 only on a base of respondents who passed all 3 attention checks. The relevant quote in the manuscript was "To replicate more directly the original conditions of Ballard and Lewandowsky (2015), we repeated the analysis with only those respondents included who passed all three initial attention checks. Table 3 shows means and ANOVA results for this restricted subset of participants, again revealing null results." We therefore concluded that attention did not serve as a hidden precondition for the effect. We

are grateful that Reviewer 1 caught a confusing sentence in the general discussion about “motivated audiences” and we eliminated this from the general discussion. We hope the clear references to the attention-checks in the materials and results section clarify these concerns.

My response: The authors appear to have misunderstood my suggestion - I apologize for my lack of clarity on this point and hope that my re-phrasing makes this recommendation clearer. By stating that the number of attention checks could be used as a moderator, I meant that the authors could include the interaction between the number of attention checks passed and the treatment variables (focus and timing) in their model to see if the number of attention checks passed significantly moderates the treatment effects. This would be a way to test whether participants who were more or less attentive responded to the treatment differently. For example, if the interaction between focus and number of attention checks passed is non-significant, it would suggest that level of attention did not affect response to the treatment. If the interaction between focus and number of attention checks passed is significant, it would suggest that it does. This analysis would not require collecting new data, but merely adding a bit to the analysis in Experiment 1, and to me it seems like another useful test of whether attention played a role in the effects (one that would complement the authors’ approach of repeating the analysis with only those respondents included who passed all three initial attention checks).

I think removing the discussion about motivated audiences renders the collection of additional data unnecessary to support the claims of the authors. The authors could instead add this as a suggestion for future research in the discussion - i.e., they tested one form of attention via attention checks in Experiment 1, but perhaps other forms of motivation/attention (e.g., need for cognition) would moderate the effects.

In addition, providing the attention checks in a supplement or online repository (e.g., the Open Science Framework) would also be useful to readers, so that they can understand what kind of attention checks were used, if they are commonplace, etc. (I was unable to access the supplemental materials, so apologies if this information is already available in the supplement!)

Comment #2: Regarding Reviewer 1’s second comment, the information sections included fewer examples of causes, to reduce burden for the respondents, but otherwise did not differ significantly from the original study. We mention this in the Materials section for Experiment 1.

My response: I appreciate this clarification. If the authors were able to make the revised (and perhaps original) materials available (e.g., on the Open Science Framework) for any of the studies, this would be highly useful for readers who might be interested in understanding these differences in greater depth. (Again, I was unable to access the supplemental materials, so apologies if this information is already available in the supplement!)

Comment #3: We addressed Reviewer 1’s third query by clarifying that the sample may have been less attentive in terms of the attention checks, but nonetheless attentive enough to participate in the study when discussing the attention variable in the Results section of Experiment 1. The present revision largely addresses the issue of attention in producing the original Ballard and Lewandowsky (2015) findings empirically, as we described above.

My response: I appreciated the authors making the effort to inquire with SSI about their sample. However, some of the justification provided was confusing to me. For example, the authors wrote on page 6 “...participants in online studies tend to read in an “F” pattern, rather than reading every sentence...” Given that they suggest that the panels used in Ballard and Lewandowsky’s original study and Experiment 1 were very similar, shouldn’t this have been true for both panels? Can there be some other reason for the very high number of participants who failed the attention checks, or does it just seem to have been luck that Ballard and Lewandowsky had a more

attentive sample for their original study than the sample in Experiment 1? Does SSI differ from Qualtrics panels like the one used in the Ballard and Lewandowsky study in some kind of important way?

Providing the attention checks used in the replication and ideally also the original studies (or mentioning if the attention checks used in the original study and replication were the same, and if not how they are different) could perhaps help to address this concern.

The authors could mention in the discussion that, given that these attention checks were particularly hard to pass in SSI's research, the group that did pass the attention checks should have been highly attentive, and yet still did not show the effect.

Comment #4: We addressed the 4th question by adding text to the "Participants" section of Experiment 1, confirming that the online panel methodology used by Ballard and Lewandowsky (2015) is similar to the methodology employed in the current Experiment 1. The terms "multiple imputation" used by the Reviewer does not appear in any of our work, nor in the original Ballard and Lewandowsky paper.

My response: The Ballard and Lewandowsky paper does indicate that the authors used imputation to replace missing values in the dataset, but the term is "hot deck imputation" rather than "multiple imputation." I apologize for indicating that the imputation was multiple imputation when it was hot deck imputation. The relevant sentence from the Ballard and Lewandowsky manuscript is at the bottom of page 4:

"Following the recommendations of Myers [21], we used hot deck imputation to deal with the missing data from the seven respondents who completed some but not all of the test items. We imputed 445 observations using the 'hot.deck' package in R [22]. After imputation, the total number of observations in the dataset was 17 444."

Thus, my question as to whether this same methodological step was followed still stands.

In general, it might be useful for readers to have a detailed summary and/or table of all of the differences between the original study and the replication in Experiment 1, so that readers can more easily compare the two studies and judge the relative strength of the evidence, given the many differences between the two studies.

Comment #5: We were unsure how to address Reviewer 1's fifth concern, regarding the potentially confusing nature of the within-subjects design. The original Ballard and Lewandowsky (2015) study that initially showed positive results, and our replication in Experiment 1, were all within-subject designs and should have been equally confusing or clear.

My response: I am confused by the authors' claim that all of the studies were within-subject designs. See the bottom of page 5 of the Ballard and Lewandowsky paper, which clearly indicates that this study had a between-subjects design:

"We used a between-groups design with two levels of the single experimental variable: outcome-uncertain versus time-uncertain presentation of uncertainty. We presented participants in each condition with information about the same four indicators of climate change: global mean surface temperature (GMST) rise, sea-level rise, ocean acidification and reduction in Arctic sea ice extent. For each indicator, we asked participants to provide information about their perceived levels of certainty and concern, perceived seriousness and endorsement of mitigative action."

It thus appears that this study was between-subjects: all participants saw only either the outcome-

uncertain condition or the time-uncertain condition and then completed the dependent variables. Experiment 1 in the current manuscript also appears to be a between-subjects design in this same way, unless I am missing something in the authors' description – it appears that participants were randomly assigned to see either outcome-uncertain information or outcome-certain information.

For Experiment 2, it appears that participants read statements in both conditions – that is, they sometimes saw outcome-uncertain statements about diseases, and sometimes saw time-uncertain statements about diseases. Thus, from my reading, this study appears to be a within-subject design, as opposed to the other two studies (Experiment 1 and the original Ballard and Lewandowsky study), because participants were exposed to both forms of uncertainty. Have I misunderstood something here? Did all participants in this study see only five statements that were outcome-uncertain, and only statements that were time-uncertain, and thus this study is between-subjects, like Experiment 1 and the original Ballard and Lewandowsky study? The authors need to more clearly and explicitly state the design(s) of the studies in the manuscript so that other readers do not become confused.

Comment #6: We added additional context and clarification about the sixth concern, about the move from climate change to disease - there was a certain amount of overlap between the execution of Experiment 1 and the design and pretesting of Experiment 2, so the narrative isn't as neatly serial as it often appears in published work.

My response: I appreciated the authors' clarification on this point, as well as their note that the manipulation should theoretically generalize to this context.

Comment #7: Finally, we believe the pretests constituted random noise and we state this in the revised manuscript.

My response: I appreciated this clarification.

Review form: Reviewer 2

Is the manuscript scientifically sound in its present form?

Yes

Are the interpretations and conclusions justified by the results?

Yes

Is the language acceptable?

Yes

Is it clear how to access all supporting data?

Yes

Do you have any ethical concerns with this paper?

No

Have you any concerns about statistical analyses in this paper?

No

Recommendation?

Accept as is

Comments to the Author(s)

The authors added statements clarifying that they intended this set of studies to be a conceptual replication of their past publication.

Decision letter (RSOS-180475.R1)

01-Feb-2019

Dear Dr Sleeth-Keppler:

Manuscript ID RSOS-180475.R1 entitled "Does "When" Really Feel More Certain than "If"?: Two failures to replicate Ballard and Lewandowsky (2015)" which you submitted to Royal Society Open Science, has been reviewed. The comments of the reviewer(s) are included at the bottom of this letter.

Please submit a copy of your revised paper before 24-Feb-2019. Please note that the revision deadline will expire at 00.00am on this date. If we do not hear from you within this time then it will be assumed that the paper has been withdrawn. In exceptional circumstances, extensions may be possible if agreed with the Editorial Office in advance. We do not allow multiple rounds of revision so we urge you to make every effort to fully address all of the comments at this stage. If deemed necessary by the Editors, your manuscript will be sent back to one or more of the original reviewers for assessment. If the original reviewers are not available we may invite new reviewers.

- Ethics statement

- Data accessibility

It is a condition of publication that all supporting data are made available either as supplementary information or preferably in a suitable permanent repository. The data

accessibility section should state where the article's supporting data can be accessed. This section should also include details, where possible of where to access other relevant research materials such as statistical tools, protocols, software etc can be accessed. If the data have been deposited in an external repository this section should list the database, accession number and link to the DOI for all data from the article that have been made publicly available. Data sets that have been deposited in an external repository and have a DOI should also be appropriately cited in the manuscript and included in the reference list.

- **Competing interests**

- **Authors' contributions**

- **Acknowledgements**

- **Funding statement**

Kind regards,

Andrew Dunn

on behalf of Dr Anastasia Christakou (Associate Editor) and Antonia Hamilton (Subject Editor)

Associate Editor Comments to Author (Dr Anastasia Christakou):

Unusually, the Editors have offered a further round of major revision for your paper. No further opportunities will be granted to revise your manuscript. Please ensure that you fully respond to the reviewers' concerns.

Reviewer comments to Author:
Reviewer: 1

Comments to the Author(s)

I have read the revised manuscript as well as the authors' responses and would like to take the opportunity to clarify a few of my remarks to the authors and to ask some outstanding questions. I include the authors' comments and my responses to their comments below.

Comment #1: Regarding Reviewer 1's first set of comments, we clarified in the Overview to Exp. 1 that no further data was collected from Ballard and Lewandowsky's original dismissed participants, making any further tests of attention as a moderator of the effects impossible. This was also a big motivation for us to conduct the present studies. Collecting more data, as Reviewer 1 suggests, would defeat the purpose of diagnosing the role of attention further, because Experiment 1 in the present manuscript was already designed to address this issue. Our original manuscript included analyses of the attention checks as variables in a multiple regression analysis (finding null results), and we also replicated the conditions of the original Ballard and Lewandowsky (2015) study by analyzing our present Experiment 1 only on a base of respondents who passed all 3 attention checks. The relevant quote in the manuscript was "To replicate more directly the original conditions of Ballard and Lewandowsky (2015), we repeated the analysis with only those respondents included who passed all three initial attention checks. Table 3 shows means and ANOVA results for this restricted subset of participants, again revealing null results." We therefore concluded that attention did not serve as a hidden precondition for the effect. We are grateful that Reviewer 1 caught a confusing sentence in the general discussion about "motivated audiences" and we eliminated this from the general discussion. We hope the clear references to the attention-checks in the materials and results section clarify these concerns.

My response: The authors appear to have misunderstood my suggestion - I apologize for my lack of clarity on this point and hope that my re-phrasing makes this recommendation clearer. By stating that the number of attention checks could be used as a moderator, I meant that the authors could include the interaction between the number of attention checks passed and the treatment variables (focus and timing) in their model to see if the number of attention checks passed significantly moderates the treatment effects. This would be a way to test whether participants who were more or less attentive responded to the treatment differently. For example, if the interaction between focus and number of attention checks passed is non-significant, it would suggest that level of attention did not affect response to the treatment. If the interaction between focus and number of attention checks passed is significant, it would suggest that it does. This analysis would not require collecting new data, but merely adding a bit to the analysis in Experiment 1, and to me it seems like another useful test of whether attention played a role in the effects (one that would complement the authors' approach of repeating the analysis with only those respondents included who passed all three initial attention checks).

I think removing the discussion about motivated audiences renders the collection of additional data unnecessary to support the claims of the authors. The authors could instead add this as a suggestion for future research in the discussion - i.e., they tested one form of attention via attention checks in Experiment 1, but perhaps other forms of motivation/attention (e.g., need for cognition) would moderate the effects.

In addition, providing the attention checks in a supplement or online repository (e.g., the Open Science Framework) would also be useful to readers, so that they can understand what kind of attention checks were used, if they are commonplace, etc. (I was unable to access the supplemental materials, so apologies if this information is already available in the supplement!)

Comment #2: Regarding Reviewer 1's second comment, the information sections included fewer examples of causes, to reduce burden for the respondents, but otherwise did not differ significantly from the original study. We mention this in the Materials section for Experiment 1.

My response: I appreciate this clarification. If the authors were able to make the revised (and perhaps original) materials available (e.g., on the Open Science Framework) for any of the studies, this would be highly useful for readers who might be interested in understanding these differences in greater depth. (Again, I was unable to access the supplemental materials, so apologies if this information is already available in the supplement!)

Comment #3: We addressed Reviewer 1's third query by clarifying that the sample may have been less attentive in terms of the attention checks, but nonetheless attentive enough to participate in the study when discussing the attention variable in the Results section of Experiment 1. The present revision largely addresses the issue of attention in producing the original Ballard and Lewandowsky (2015) findings empirically, as we described above.

My response: I appreciated the authors making the effort to inquire with SSI about their sample. However, some of the justification provided was confusing to me. For example, the authors wrote on page 6 "...participants in online studies tend to read in an "F" pattern, rather than reading every sentence..." Given that they suggest that the panels used in Ballard and Lewandowsky's original study and Experiment 1 were very similar, shouldn't this have been true for both panels? Can there be some other reason for the very high number of participants who failed the attention checks, or does it just seem to have been luck that Ballard and Lewandowsky had a more attentive sample for their original study than the sample in Experiment 1? Does SSI differ from Qualtrics panels like the one used in the Ballard and Lewandowsky study in some kind of important way?

Providing the attention checks used in the replication and ideally also the original studies (or mentioning if the attention checks used in the original study and replication were the same, and if not how they are different) could perhaps help to address this concern.

The authors could mention in the discussion that, given that these attention checks were particularly hard to pass in SSI's research, the group that did pass the attention checks should have been highly attentive, and yet still did not show the effect.

Comment #4: We addressed the 4th question by adding text to the "Participants" section of Experiment 1, confirming that the online panel methodology used by Ballard and Lewandowsky (2015) is similar to the methodology employed in the current Experiment 1. The terms "multiple imputation" used by the Reviewer does not appear in any of our work, nor in the original Ballard and Lewandowsky paper.

My response: The Ballard and Lewandowsky paper does indicate that the authors used imputation to replace missing values in the dataset, but the term is "hot deck imputation" rather than "multiple imputation." I apologize for indicating that the imputation was multiple imputation when it was hot deck imputation. The relevant sentence from the Ballard and Lewandowsky manuscript is at the bottom of page 4:

"Following the recommendations of Myers [21], we used hot deck imputation to deal with the missing data from the seven respondents who completed some but not all of the test items. We imputed 445 observations using the 'hot.deck' package in R [22]. After imputation, the total number of observations in the dataset was 17 444."

Thus, my question as to whether this same methodological step was followed still stands.

In general, it might be useful for readers to have a detailed summary and/or table of all of the differences between the original study and the replication in Experiment 1, so that readers can more easily compare the two studies and judge the relative strength of the evidence, given the many differences between the two studies.

Comment #5: We were unsure how to address Reviewer 1's fifth concern, regarding the potentially confusing nature of the within-subjects design. The original Ballard and Lewandowsky (2015) study that initially showed positive results, and our replication in Experiment 1, were all within-subject designs and should have been equally confusing or clear.

My response: I am confused by the authors' claim that all of the studies were within-subject designs. See the bottom of page 5 of the Ballard and Lewandowsky paper, which clearly indicates that this study had a between-subjects design:

"We used a between-groups design with two levels of the single experimental variable: outcome-uncertain versus time-uncertain presentation of uncertainty. We presented participants in each condition with information about the same four indicators of climate change: global mean surface temperature (GMST) rise, sea-level rise, ocean acidification and reduction in Arctic sea ice extent. For each indicator, we asked participants to provide information about their perceived levels of certainty and concern, perceived seriousness and endorsement of mitigative action."

It thus appears that this study was between-subjects: all participants saw only either the outcome-uncertain condition or the time-uncertain condition and then completed the dependent variables. Experiment 1 in the current manuscript also appears to be a between-subjects design in this same way, unless I am missing something in the authors' description – it appears that participants were randomly assigned to see either outcome-uncertain information or outcome-certain information.

For Experiment 2, it appears that participants read statements in both conditions – that is, they sometimes saw outcome-uncertain statements about diseases, and sometimes saw time-uncertain statements about diseases. Thus, from my reading, this study appears to be a within-subject design, as opposed to the other two studies (Experiment 1 and the original Ballard and Lewandowsky study), because participants were exposed to both forms of uncertainty. Have I misunderstood something here? Did all participants in this study see only five statements that were outcome-uncertain, and only statements that were time-uncertain, and thus this study is between-subjects, like Experiment 1 and the original Ballard and Lewandowsky study? The authors need to more clearly and explicitly state the design(s) of the studies in the manuscript so that other readers do not become confused.

Comment #6: We added additional context and clarification about the sixth concern, about the move from climate change to disease - there was a certain amount of overlap between the execution of Experiment 1 and the design and pretesting of Experiment 2, so the narrative isn't as neatly serial as it often appears in published work.

My response: I appreciated the authors' clarification on this point, as well as their note that the manipulation should theoretically generalize to this context.

Comment #7: Finally, we believe the pretests constituted random noise and we state this in the revised manuscript.

My response: I appreciated this clarification.

Reviewer: 2

Comments to the Author(s)

The authors added statements clarifying that they intended this set of studies to be a conceptual replication of their past publication.

Author's Response to Decision Letter for (RSOS-180475.R1)

See Appendix B.

RSOS-180475.R2 (Revision)

Review form: Reviewer 2

Is the manuscript scientifically sound in its present form?

Yes

Are the interpretations and conclusions justified by the results?

No

Is the language acceptable?

Yes

Is it clear how to access all supporting data?

Yes

Do you have any ethical concerns with this paper?

No

Have you any concerns about statistical analyses in this paper?

No

Recommendation?

Reject

Comments to the Author(s)

My comments remain the same as in previous rounds of review. I defer to the editor for decision making.

Decision letter (RSOS-180475.R2)

17-Jun-2019

Dear Dr Sleeth-Keppler,

I am pleased to inform you that your manuscript entitled "Does "When" Really Feel More Certain than "If"?: Two failures to replicate Ballard and Lewandowsky (2015)" is now accepted for publication in Royal Society Open Science.

on behalf of Dr Anastasia Christakou (Associate Editor) and Antonia Hamilton (Subject Editor)
openscience@royalsociety.org

Associate Editor Comments to Author (Dr Anastasia Christakou):

Associate Editor: 1

Comments to the Author:

Please accept our apologies for the unusual time this revision has been under review. We've received further comments from one of the reviewers of the previous two versions of your paper. They have reviewed your responses to their own comments and those of a second referee from the prior version, who was regrettably unavailable to assess this round.

The Editors have come to the conclusion that the paper may be accepted 'as is': the referee deferred to the Editors' judgement, and on reviewing the previous iterations of the paper, and your responses to earlier comments, the Editors consider the paper is now ready for acceptance -- any further 'review' of this attempt at a replication, and remaining concerns referees or readers may have, should take place in the public domain, which can be best served by publishing the paper. The authors might like to bear in mind that the journal encourages formal replications and, indeed, launched an article type to support this process - should the authors be considering any further replication studies, we would encourage you to submit to the journal via the replication track.

Reviewer comments to Author:

Reviewer: 2

Comments to the Author(s)

My comments remain the same as in previous rounds of review. I defer to the editor for decision making.

Appendix A

Dear Dr Anastasia Christakou (Associate Editor) and Prof. Antonia Hamilton (Subject Editor)

Thank you for the opportunity to revise our manuscript. To start, we addressed the second Reviewer's comments by clarifying in several places that the studies we report are conceptual, not exact replications. Reviewer 1's first block of concerns all involved the role of attention in the initial Ballard and Lewandowsky (2015) work - specific comments to this effects are on pg. 3, final paragraph of the introduction and throughout the manuscript. In short, we embarked on the replications reported in this manuscript precisely because Ballard and Lewandowsky had dismissed their initial pool of participants who had failed the attention checks, and they were unable to test for statistical effects of attention on the outcomes.

- Regarding Reviewer 1's first set of comments, we clarified in the Overview to Exp. 1 that no further data was collected from Ballard and Lewandowsky's original dismissed participants, making any further tests of attention as a moderator of the effects impossible. This was also a big motivation for us to conduct the present studies. Collecting more data, as Reviewer 1 suggests, would defeat the purpose of diagnosing the role of attention further, because Experiment 1 in the present manuscript was already designed to address this issue.
- Our original manuscript included analyses of the attention checks as variables in a multiple regression analysis (finding null results), and we also replicated the conditions of the original Ballard and Lewandowsky (2015) study by analyzing our present Experiment 1 only on a base of respondents who passed all 3 attention checks. The relevant quote in the manuscript was "To replicate more directly the original conditions of Ballard and Lewandowsky (2015), we repeated the analysis with only those respondents included who passed all three initial attention checks. Table 3 shows means and ANOVA results for this restricted subset of participants, again revealing null results. " We therefore concluded that attention did not serve as a hidden precondition for the effect. We are grateful that Reviewer 1 caught a confusing sentence in the general discussion about "motivated audiences" and we eliminated this from the general discussion. We hope the clear references to the attention-checks in the materials and results section clarify these concerns.
- Regarding Reviewer 1's second comment, the information sections included fewer examples of causes, to reduce burden for the respondents, but otherwise did not differ significantly from the original study. We mention this in the Materials section for Experiment 1.
- We addressed Reviewer 1's third query by clarifying that the sample may have been less attentive in terms of the attention checks, but nonetheless attentive enough to participate in the study when discussing the attention variable in the Results section of Experiment 1. The present revision largely addresses the issue of attention in producing the original Ballard and Lewandowsky (2015) findings empirically, as we described above.
- We addressed the 4th question by adding text to the "Participants" section of Experiment 1, confirming that the online panel methodology used by Ballard and Lewandowsky (2015) is similar to the methodology employed in the current Experiment 1. The terms

“multiple imputation” used by the Reviewer does not appear in any of our work, nor in the original Ballard and Lewandowsky paper.

- We were unsure how to address Reviewer 1’s fifth concern, regarding the potentially confusing nature of the within-subjects design. The original Ballard and Lewandowsky (2015) study that initially showed positive results, and our replication in Experiment 1, were all within-subject designs and should have been equally confusing or clear.
- We added additional context and clarification about the sixth concern, about the move from climate change to disease - there was a certain amount of overlap between the execution of Experiment 1 and the design and pretesting of Experiment 2, so the narrative isn’t as neatly serial as it often appears in published work.
- Finally, we believe the pretests constituted random noise and we state this in the revised manuscript.

Appendix B

Dear Dr Anastasia Christakou (Associate Editor) and Prof. Antonia Hamilton (Subject Editor)

Thank you for the opportunity to revise our manuscript a second time. We are confident this final revision addresses all of Reviewer 1's concerns. Below, we include Reviewer 1's most recent comments to us, and our replies for each.

Reviewer 1: The authors appear to have misunderstood my suggestion - I apologize for my lack of clarity on this point and hope that my re-phrasing makes this recommendation clearer. By stating that the number of attention checks could be used as a moderator, I meant that the authors could include the interaction between the number of attention checks passed and the treatment variables (focus and timing) in their model to see if the number of attention checks passed significantly moderates the treatment effects. This would be a way to test whether participants who were more or less attentive responded to the treatment differently. For example, if the interaction between focus and number of attention checks passed is non-significant, it would suggest that level of attention did not affect response to the treatment. If the interaction between focus and number of attention checks passed is significant, it would suggest that it does. This analysis would not require collecting new data, but merely adding a bit to the analysis in Experiment 1, and to me it seems like another useful test of whether attention played a role in the effects (one that would complement the authors' approach of repeating the analysis with only those respondents included who passed all three initial attention checks).

I think removing the discussion about motivated audiences renders the collection of additional data unnecessary to support the claims of the authors. The authors could instead add this as a suggestion for future research in the discussion – i.e., they tested one form of attention via attention checks in Experiment 1, but perhaps other forms of motivation/attention (e.g., need for cognition) would moderate the effects.

In addition, providing the attention checks in a supplement or online repository (e.g., the Open Science Framework) would also be useful to readers, so that they can understand what kind of attention checks were used, if they are commonplace, etc. (I was unable to access the supplemental materials, so apologies if this information is already available in the supplement!)

Our Response: 1) We appreciate the reviewer's clarification regarding their recommendation for the attention checks analysis. We conducted the analysis recommended by separately testing the focus*attention, timing*attention, and focus*timing*attention interactions for both the certainty and concern outcomes. In no case was the interaction significant. We have added a footnote to the manuscript describing the test and outcome; 2) we added a sentence about the possibility for future research in the discussion for experiment 1; 3) we added the attention checks from the original Ballard and Lewandowsky (2015) study and the checks used in Experiment 1 as an online supplement. Importantly, based on a detailed comparison of the attention filters, we concluded that differences in the questions used, rather than sample differences, explain the different failure rates around attention-focus. We included additional comments to this effect in the Results section for experiment 1.

My response: I appreciate this clarification. If the authors were able to make the revised (and perhaps original) materials available (e.g., on the Open Science Framework) for any of the studies, this would be highly useful for readers who might be interested in understanding these

differences in greater depth. (Again, I was unable to access the supplemental materials, so apologies if this information is already available in the supplement!)

Our Response: We added the entire programmed survey for Experiment 1 as an online supplemental material. The programmed survey for Ballard and Lewandowsky's study is still available as an online supplement along with the original paper. To avoid confusion in navigating multiple online materials for studies published in different journals, we chose to only publish the materials for our present Experiment 1 on climate change effects.

Reviewer 1: I appreciated the authors making the effort to inquire with SSI about their sample. However, some of the justification provided was confusing to me. For example, the authors wrote on page 6 "...participants in online studies tend to read in an "F" pattern, rather than reading every sentence..." Given that they suggest that the panels used in Ballard and Lewandowsky's original study and Experiment 1 were very similar, shouldn't this have been true for both panels? Can there be some other reason for the very high number of participants who failed the attention checks, or does it just seem to have been luck that Ballard and Lewandowsky had a more attentive sample for their original study than the sample in Experiment 1? Does SSI differ from Qualtrics panels like the one used in the Ballard and Lewandowsky study in some kind of important way?

Providing the attention checks used in the replication and ideally also the original studies (or mentioning if the attention checks used in the original study and replication were the same, and if not how they are different) could perhaps help to address this concern.

The authors could mention in the discussion that, given that these attention checks were particularly hard to pass in SSI's research, the group that did pass the attention checks should have been highly attentive, and yet still did not show the effect.

Authors' Response: 1) We are including the original attention checks used by Ballard and Lewandowsky (2015) in the online supplemental materials section. 2) We added a relevant sentence to the Results section, and discussion about attention. Comparing the attention checks to the ones used by Ballard and Lewandowsky, we confidently concluded that the differences in the difficulty of the attention filters contributed to the high failure rate in Experiment 1.

Reviewer 1: The Ballard and Lewandowsky paper does indicate that the authors used imputation to replace missing values in the dataset, but the term is "hot deck imputation" rather than "multiple imputation." I apologize for indicating that the imputation was multiple imputation when it was hot deck imputation. The relevant sentence from the Ballard and Lewandowsky manuscript is at the bottom of page 4:

"Following the recommendations of Myers [21], we used hot deck imputation to deal with the missing data from the seven respondents who completed some but not all of the test items. We imputed 445 observations using the 'hot.deck' package in R [22]. After imputation, the total number of observations in the dataset was 17 444."

Thus, my question as to whether this same methodological step was followed still stands.

In general, it might be useful for readers to have a detailed summary and/or table of all of the differences between the original study and the replication in Experiment 1, so that readers can more easily compare the two studies and judge the relative strength of the evidence, given the many differences between the two studies.

Our Response: We added the requested table as table number 5 in the discussion section.

My response: I am confused by the authors' claim that all of the studies were within-subject designs. See the bottom of page 5 of the Ballard and Lewandowsky paper, which clearly indicates that this study had a between-subjects design:

"We used a between-groups design with two levels of the single experimental variable: outcome-uncertain versus time-uncertain presentation of uncertainty. We presented participants in each condition with information about the same four indicators of climate change: global mean surface temperature (GMST) rise, sea-level rise, ocean acidification and reduction in Arctic sea ice extent. For each indicator, we asked participants to provide information about their perceived levels of certainty and concern, perceived seriousness and endorsement of mitigative action."

It thus appears that this study was between-subjects: all participants saw only either the outcome-uncertain condition or the time-uncertain condition and then completed the dependent variables. Experiment 1 in the current manuscript also appears to be a between-subjects design in this same way, unless I am missing something in the authors' description – it appears that participants were randomly assigned to see either outcome-uncertain information or outcome-certain information.

For Experiment 2, it appears that participants read statements in both conditions – that is, they sometimes saw outcome-uncertain statements about diseases, and sometimes saw time-uncertain statements about diseases. Thus, from my reading, this study appears to be a within-subject design, as opposed to the other two studies (Experiment 1 and the original Ballard and Lewandowsky study), because participants were exposed to both forms of uncertainty. Have I misunderstood something here? Did all participants in this study see only five statements that were outcome-uncertain, and only statements that were time-uncertain, and thus this study is between-subjects, like Experiment 1 and the original Ballard and Lewandowsky study? The authors need to more clearly and explicitly state the design(s) of the studies in the manuscript so that other readers do not become confused.

Our Response: We apologize for the confusion; the second experiment was indeed a within-subjects design, with randomization of the order of the diseases and randomization of projection formats for each disease. We added a simple statement to the methods to clarify.

Reviewer 1: I appreciated the authors' clarification on this point, as well as their note that the manipulation should theoretically generalize to this context.

Comment #7: Finally, we believe the pretests constituted random noise and we state this in the revised manuscript.

Reviewer 1: I appreciated this clarification.